# Interleukin-11/IL-11 Receptor Promotes Glioblastoma Cell Proliferation, Epithelial–Mesenchymal Transition, and Invasion

**DOI:** 10.3390/brainsci14010089

**Published:** 2024-01-17

**Authors:** Sarah F. Stuart, Peter Curpen, Adele J. Gomes, Michelle C. Lan, Shuai Nie, Nicholas A. Williamson, George Kannourakis, Andrew P. Morokoff, Adrian A. Achuthan, Rodney B. Luwor

**Affiliations:** 1Department of Surgery, The University of Melbourne, The Royal Melbourne Hospital, Parkville, VIC 3050, Australia; sstuart@student.unimelb.edu.au (S.F.S.); adelejoyceg@student.unimelb.edu.au (A.J.G.); morokoff@unimelb.edu.au (A.P.M.); 2Fiona Elsey Cancer Research Institute, Ballarat, VIC 3350, Australia; george@fecri.org.au; 3Townsville Hospital and Health Service, James Cook University, Townsville, QLD 4814, Australia; jensencurpen@outlook.com; 4Melbourne Mass Spectrometry and Proteomics Facility, Bio21 Molecular Science & Biotechnology Institute, The University of Melbourne, Melbourne, VIC 3052, Australia; shuai.nie@unimelb.edu.au (S.N.); nawill@unimelb.edu.au (N.A.W.); 5Federation University, Ballarat, VIC 3350, Australia; 6Department of Neurosurgery, The Royal Melbourne Hospital, Parkville, VIC 3050, Australia; 7Department of Medicine, The University of Melbourne, The Royal Melbourne Hospital, Parkville, VIC 3050, Australia; aaa@unimelb.edu.au

**Keywords:** interleukin 11, glioblastoma, EMT, invasion

## Abstract

Glioblastoma is highly proliferative and invasive. However, the regulatory cytokine networks that promote glioblastoma cell proliferation and invasion into other areas of the brain are not fully defined. In the present study, we define a critical role for the IL-11/IL-11Rα signalling axis in glioblastoma proliferation, epithelial to mesenchymal transition, and invasion. We identified enhanced IL-11/IL-11Rα expression correlated with reduced overall survival in glioblastoma patients using TCGA datasets. Proteomic analysis of glioblastoma cell lines overexpressing IL-11Rα displayed a proteome that favoured enhanced proliferation and invasion. These cells also displayed greater proliferation and migration, while the knockdown of IL-11Rα reversed these tumourigenic characteristics. In addition, these IL-11Rα overexpressing cells displayed enhanced invasion in transwell invasion assays and in 3D spheroid invasion assays, while knockdown of IL-11Rα resulted in reduced invasion. Furthermore, IL-11Rα-overexpressing cells displayed a more mesenchymal-like phenotype compared to parental cells and expressed greater levels of the mesenchymal marker Vimentin. Overall, our study identified that the IL-11/IL-11Rα pathway promotes glioblastoma cell proliferation, EMT, and invasion.

## 1. Introduction

Glioblastoma is the most frequent and aggressive malignant brain tumour in adults, characterised by its neurologically destructive and infiltrative nature [1,2,3,4,5,6]. Glioblastoma was further classified by the WHO in 2021 as exclusively isocitrate dehydrogenase (IDH)-wildtype, with necrosis or microvascular proliferation, or mutations in the *TERT* promoter, chromosomes +7/−10, and the EGFR [6,7]. While glioblastoma tumour cells are believed to be derived from mutations in structural cells of the brain called astrocytes, the exact cause of this disease remains unknown [8]. Glioblastoma makes up 15.4% of all primary brain tumours and more than 50% of all malignant brain tumours in adults [9]. According to the WHO, the incidence of glioblastoma is between 3 and 4 people per 100,000 per year, or approximately 17,000 cases annually [10,11,12,13]. Although this number is dramatically lower than many other cancers, this disease is responsible for 2.5% of all cancer-related deaths [13].

The prognosis of glioblastoma is very poor, with a median overall survival of 12–15 months after diagnosis [14,15,16,17]. This is mainly due to the invariable tumour recurrence arising in almost all glioblastoma patients [3,18,19]. Five-year survival is approximately 5%, and ‘long-term survivors’ are classified as those who live longer than two years after diagnosis [20,21,22,23]. It has also been shown that suicidal ideation and attempts increase in glioblastoma patients compared to the number of attempts in the general population [24].

However, the key molecular drivers that promote this aggressiveness are not completely known or understood. The tumour microenvironment contains many pro-tumourigenic cytokines and growth factors and is now recognised as a major participant in promoting glioblastoma development and progression. IL-11 is a member of the IL-6 cytokine superfamily and is a pleiotropic cytokine that binds its specific receptor (IL-11Rα) and the transmembrane co-receptor GP130 [25,26]. The formation of this heterotrimer activates JAK proteins and, in turn, leads to the activation of the transcription factor STAT3 [27,28]. STAT3 is found to be over-activated in several malignancies and mediates many oncogenic processes, including proliferation, migration, and invasion [27,29]. Although most research into STAT3 activation and downstream signalling has mainly been studied in the context of IL-6 stimulation, recent studies have shown an important role for IL-11 in STAT3 activation, especially in the initiation and progression of cancer [30,31,32,33]. In fact, IL-11 has a higher correlation with STAT3 activation than the prototypical family member, IL-6 [34]. IL-11 has now been identified as a driver in a wide range of malignancies, including breast, prostate, endometrial, ovarian, liver, and gastric cancers, and is considered an important biomarker in determining the prognosis of patients with breast, lung, and gastric cancers [30,31,32,33]. Specifically, IL-11 has been shown to increase tumour proliferation, migration, invasion, and survival, all important hallmarks of cancer [30,31,32,33]. As a result of these findings, human monoclonal antibodies have been developed against IL-11 with success in treating some cancers, including endometrial cancer [35]. Furthermore, IL-11 receptor inhibition has shown little off-target effects, contrasting the detrimental side effects associated with JAK inhibitors or the low efficacy of STAT inhibitors.

Despite the growing evidence of the importance of IL-11 in these tumour types, the current knowledge regarding the literature on the role of IL-11 in glioblastoma progression is very limited. Recently, we demonstrated the role of IL-11/IL-11Rα in promoting glioblastoma metabolic adaptation, leading to enhanced survival [36]. Another study determined that IL-11 is secreted by glioblastoma cells. However, the potential functional role of IL-11 in enhancing glioblastoma development and progression was not clearly determined [37,38]. Therefore, here we determined the role of IL-11 signalling in glioblastoma tumour progression. Specifically, we explored a potential correlation between IL-11 and/or IL-11Rα expression in glioblastoma versus normal brain tissue and glioblastoma patient survival. We then explore the biological effects of both overexpression and knockdown of IL-11Rα expression on glioblastoma proliferation, migration, and invasion in both two- and three-dimensional assays.

## 2. Materials and Methods

Antibodies and Reagents: The rabbit polyclonal antibodies directed against pSTAT3, STAT3, and GAPDH were all obtained from Cell Signalling Technology (Danvers, MA, USA). Human IL-11Rα and negative control siRNA were from Thermofisher Scientific (Scoresby, VIC, Australia).

Cell Culture: The primary glioblastoma cell lines #20 and #28 were originally derived from two patients with pathologically confirmed glioblastoma at the Royal Melbourne Hospital and subsequently modified from neurosphere non-adherent cells to adherent cells grown in monolayer by disassociating spheroid cultures and seeding cells onto adherent plates. The use of these cell lines in the laboratory was approved by the Melbourne Health Human Research and Ethics Committee (HREC 2012.219). All cell lines were maintained in Dulbecco’s Modified Eagle’s Medium (Life Technologies, Carlsbad, CA, USA) that contained 5% (*v*/*v*) foetal bovine serum (FBS) (Life Technologies), 100 U/mL penicillin, and 100 µg/mL streptomycin (Life Technologies). Transient transfection was performed using Metafectene Pro (Biontex; München, Germany), as per the manufacturer’s instructions, with control or IL-11Rα siRNA. The #20 and #28 IL-11Rα stably transfected clones were generated as previously described [36]. All cells were incubated in a humidified atmosphere of 90% air and 5% CO_2_ at 37 °C. All media with variations in glucose and glutamine concentrations were purchased from Life Technologies.

Cell Viability Assays: Cells were seeded in 24-well plates and allowed to adhere overnight. After 72 h, cells were washed, and a mixture of 6.0% glutaraldehyde and 0.5% crystal violet was added for 30 min, followed by another wash, and then allowed to dry overnight. The colonies were quantified using the ImageJ Plugin [39].

Western blotting: Cells were lysed in lysis buffer (50 mM Tris (pH 7.4), 150 mM NaCl, 1% (*v*/*v*) Triton-X-100, 50 mM NaF, 2 mM MgCl_2_, 1 mM Na_3_VO_4_, and protease inhibitor cocktail (Roche; Basel, Switzerland)) and clarified by centrifugation (13,000× *g* for 15 min at 4 °C). Proteins were then separated by SDS-PAGE (Life Technologies), blotted onto nitrocellulose, and probed with the indicated primary antibodies. The signal was visualised using an enhanced chemiluminescence detection kit (GE Healthcare; Chicago, IL, USA) following incubation with appropriate secondary antibodies (Biorad Laboratories; Hercules, CA, USA). 

STAT3 Luciferase Activity: As previously described [40], cells were infected with the adenoviral STAT3 reporter (*Ad-APRE-luc*) and allowed to adhere overnight. After 24 h, cells were stimulated with ±IL-11 (100 ng/mL) in serum-free media for a further 24 h. Post 24 h, cells were lysed and assessed for STAT3 luciferase activity with the use of the Luciferase Reporter Assay Kit (Promega, Madison, WI, USA) following the manufacturer’s instructions.

RNA Extraction and RT-PCR: Cells were seeded in 6-well plates and allowed to adhere overnight. Following cell treatments and/or transfections, total RNA was extracted using an RNeasy Mini Kit (Qiagen; Hilden, Germany) following the manufacturer’s instructions. Reverse transcription was performed using the High-Capacity RNA-to-cDNA Kit (Applied Biosystems; Waltham, MA, USA). Reverse transcription-PCR was performed using the GeneAmp PCR System 2400 (Perkin Elmer, Waltham, MA, USA) under the conditions of 37 °C for 60 min and 95 °C for 5 min. To quantify the transcripts of the genes of interest, real-time PCR was performed using the ViiA 7 Real-Time PCR system (Applied Biosystems) for IL-11Rα (Applied Biosystems, Hs00234415_m1), Vimentin (Applied Biosystems, Hs00958111_m1), Snail (Snai1; Applied Biosystems, Hs00195591), and GAPDH (Applied Biosystems, Hs02758991_g1). Amplified RNA samples were calculated using the 2^−∆∆CT^ method [41]. 

Scratch/Wound Healing Assay: Cells were seeded in 6-well plates and grown to 100% confluency. After which, a scratch/wound was created on the bottom of each well using a p200 sterile tip. Cells were then cultured in media containing mitomycin C (Sigma, Darmstadt, Germany) to stop proliferation, and phase-contrast images were acquired at 0 and 24 h post-scratch. An inverted microscope (IX50 Olympus, Shinjuku City, Tokyo) and the Leica Application Suite (LAS v4.5) were used to process and capture images. ImageJ was utilised to quantify wound closure. 

Migration and Invasion Transwell Assays: Cells were seeded onto the micropore filter of the top chamber of 24-well transwell plates (Corning, Glendale, AZ, USA) in serum-free media. For invasion assays, the micropore filter was pre-coated with 10% (*v*/*v*) Matrigel 24 h prior. After 24 h, the media was removed from the upper well, and cells were incubated in formalin (5 min), crystal violet dye (5 min), and then water (5 min). Cells remaining on the upper side of the micropore filter were removed using a cotton bud, and the micropore filter was then mounted on a microscope slide. The slides were imaged at 200× magnification, and the images were analysed using Image Color Summarizer software to determine the percent of cells that had migrated/invaded through the micropore filter. 

Spheroid Invasion Assay: Cells were seeded at a concentration of 5 × 10^4^ cells in DMEM (0.5 mL) in ultra-low attachment surface 24-well plates (Corning) and left to form spheres in an incubator at 37 °C with 10% CO_2_. Multi-cellular spheres (25 μL) were mixed with Matrigel (25 μL) and seeded onto a 96-well plate. Spheres were imaged over a 24 h period, and invasion was analysed using Image J software to determine the relative growth (fold change) of the cell lines over time.

Online Database Analysis: TCGA gene expression data were obtained by using the OncoLnc database (www.oncolnc.org, accessed on 19 July 2022). For a given gene, the gene ID was entered, and ‘GBM’ was selected. Patients belonging to either the lower or upper 25th percentiles were chosen for the analysis. The Human Protein Atlas (HPA) (https://www.proteinatlas.org, accessed on 19 July 2022) was used to retrieve the protein expression level of IL-11Ra from the GBM tumour tissues and correlative normal tissues of the HPA dataset. We inputted IL-11 and IL-11Ra into cBioPortal (https://www.cbioportal.org, accessed on 19 July 2022) to obtain their genetic alteration data using the TCGA and CPTAC 2021 datasets. The STRING database (https://string-db.org, accessed on 19 July 2022) was employed for the retrieval of interacting genes/proteins on STRING to determine the top 20 proteins interacting with IL11 and IL-11Ra and construct the protein-protein interaction (PPI) network. Subsequently, we built a well-annotated, comprehensive, and publishable PPI network using Cytoscape software (https://cytoscape.org, accessed on 19 July 2022). We performed an enrichment analysis of the top 20 interacting proteins of IL-11 and IL-11Ra using the DAVID database (https://david.ncifcrf.gov, accessed on 19 July 2022)—an online software platform that integrates both Gene Ontology (GO) and KEGG analyses. By performing GO analysis, we divided differentially expressed genes (DEGs) into three major categories based on their functions, namely biological process (BP), cellular component (CC), and molecular function (MF). The KEGG database was used to understand high-level gene functions and their utilities within a specific biological system. GEPIA2 (http://gepia2.cancer-pku.cn/#index, accessed on 19 July 2022) was used to construct box plots to demonstrate the overexpression of IL-11 and IL-11Ra with a cut-off *p*-value of 0.01 and a jitter size of 0.4. In addition, survival curves for IL-11 and IL-11Ra among glioblastoma patients were plotted using the GEPIA2 online survival analysis tool and grouped based on the median. Finally, we calculated hazard ratios using Cox proportional hazards, with axis units as months and 95% confidence intervals drawn as dashed lines. A *p*-value > 0.05 was statistically significant.

Proteomics Analysis: Cells were seeded into 20 cm dishes and grown in glucose-free/serum-free DMEM for 24 h. Cell lysis buffer (50 mM Tris (pH 7.4), 150 mM NaCl, 1% Triton-X-100, 50 mM NaF, 2 mM MgCl_2_, 1 mM Na_3_VO_4_, and protease inhibitor cocktail (Roche; Basel, Switzerland)) was added to pelleted cells to yield a final volume of 23 µL containing 100 µg of protein. Cell lysates were reduced using TCEP (500 mM) and incubated for 15 min at 55 °C. Alkylate disulfides (50 mM) were added to lysates before incubation for 30 min at 45 °C. An acidifier was added to the lysates to lower the pH to <1. Wash buffer (165 µL) was added to each sample, and each mix was added to an S-trap micro spin column (Protifi). S-trap columns were centrifuged for 30 s at 4000× *g*. Protein was cleaned with three washes of the S-trap in the wash buffer. Protein was digested using trypsin (1 µg/100 µg of protein) and incubated overnight at 37 °C. Peptides were eluted for analysis before being loaded into the mass spectrometer. Samples were loaded onto a 100 µm, 2 cm nanoviper Pepmap100 trap column, eluted, and separated on an RSLC nano column 75 µm × 50 cm, Pepmap100 C18 analytical column (Thermo Fisher Scientific, Waltham, MA, USA). Peptide samples were analysed by LCMS/MS using an Orbitrap mass spectrometer (Thermo Fisher Scientific) coupled online to an RSLC nano HPLC (Ultimate 3000 UHPLC, Thermo Fisher Scientific). All generated files were analysed with MaxQuant (version 1.5.3.30) 69 and its implemented Andromeda search engine to obtain protein identifications as well as their label-free quantitation intensities. The MaxQuant result output was further processed with Perseus (Version 1.5.0.40), a module from the MaxQuant suite. Proteins with values of ≥3-fold change or ≤3-fold change between groups were selected.

Statistical Analysis: The statistical analyses for all experiments were conducted with an unpaired, two-tailed Student’s *t*-test to assess significance, and a minimum threshold of *p* < 0.05 was chosen to determine significance. The survival analyses from OncoLnc used a log-rank *t*-test to determine significance, and data were displayed on a Kaplan–Meier plot.

## 3. Results

### 3.1. IL-11Rα Promotes a Pro-Tumourigenic Protein Signature

Our previously published data have shown that IL-11 expression correlates with glioblastoma patient survival and that IL-11Rα expression promotes enhanced survival of glioblastoma cells [36]. Here, we first extended this analysis using additional publicly available databases, including the TCGA database. Both IL-11 and IL-11Rα gene expression were detected at significantly higher levels in glioblastoma patient tissue compared to normal brain tissue (Figure 1A,B), while the protein expression of IL-11Rα was found to be higher in glioblastoma tissue compared to normal tissue using the Human Protein Atlas database (Figure 1C). Both IL-11 and Il-11Rα gene expression significantly correlated with reduced overall survival in glioblastoma patients (Figure 1D,E), despite neither of these genes having a large alteration rate (Appendix A). To further elucidate the role of IL-11 signalling, we next utilised two primary glioblastoma cell lines that we had previously stably transfected with IL-11Rα [36]. These cells (designed #20-IL-11Rα and #28-IL-11Rα) displayed greater phosphorylated STAT3 and transcriptional STAT3 activity following IL-11 stimulation compared to their parental counterparts (Figure 1F,G). We next performed proteomics analysis to determine a global differential protein expression between these IL-11Rα transfected glioblastoma cell lines and their parental counterparts (Appendix A).

### 3.2. IL-11Rα Promotes Glioblastoma Cell Proliferation

As many of the differentially expressed proteins from our proteomics analysis yielded pro-tumourigenic drivers, we next specifically evaluated whether IL-11Rα expression promoted glioblastoma progression, including characteristics such as enhanced proliferation, migration, invasion, and pro-EMT properties. Both the IL-11R11Rα transfected cell lines (#20-IL-11Rα and #28-IL-11Rα) demonstrated significantly greater proliferation (Figure 2A,B) than their parental counterparts. While knockdown of IL-11Rα (as verified by PCR; Appendix A) reduced this enhanced proliferation (Figure 2C,D).

### 3.3. IL-11Rα Promotes Glioblastoma Cell Migration and Invasion

As IL-11Rα could confer enhanced proliferation, we next examined if IL-11Rα could also promote glioblastoma cell migration and invasion. Indeed, IL-11Rα transfected cells displayed greater wound healing (Figure 3A,B) and transwell migration (Figure 3C,D) compared to their parental counterparts. Knockdown of IL-11Rα reversed this effect, reducing IL-11Rα-mediated migration (Figure 3E,F).

We next evaluated the effect of IL-11Rα on cell invasion. IL-11Rα transfected cells displayed greater transwell invasion compared to their parental counterparts (Figure 4A,B), while knockdown of IL-11Rα reduced this enhanced invasion (Figure 4C,D). In addition, IL-11Rα enhanced invasion when cells were grown in three-dimensional spheroids (Figure 4E) compared to their parental counterparts.

### 3.4. IL-11Rα Promotes an EMT-like Phenotype

As IL-11Rα promoted invasion, we next determined if it could also enhance pro-EMT-like properties. As both 20-IL-11Rα and 28-IL-11Rα cells displayed a more mesenchymal-like morphology visually compared to their parental counterparts (Figure 5A), we next determined if both mesenchymal markers, Vimentin and Snail, were elevated in the IL-11Rα transfected cells. Both IL-11Rα transfected cells contained greater gene expression levels of the mesenchymal markers Vimentin and SNAIL compared to parental cells (Figure 5B,C).

## 4. Discussion

An improved understanding of the molecular mechanisms that drive glioblastoma progression will allow for improved targeted therapeutic approaches to combat this currently incurable disease. Over the last few decades, IL-11 signalling has been recognised as a key pro-tumourigenic molecule [29,42], with evidence of its importance in the progression of several cancers, including non-small cell lung cancer, pancreatic cancer, renal cell carcinoma, and breast cancer [43,44,45,46]. Here we show a clear role of IL-11/IL-11Rα signalling in glioblastoma progression. Our current data mining of the TCGA demonstrated that IL-11 and the IL-11Rα expression negatively correlated with glioblastoma patient survival. These findings, together with our previous work [36], demonstrate that IL-11 expression is the only IL-6 family member that is correlated with glioblastoma patient survival. In addition, IL-11Rα has been identified as a prognostic marker correlating negatively with patient survival and a potential candidate for drug intervention in osteosarcoma [47,48,49]. Our current data show that consistent, stable transfection of IL-11Rα in cells that have low levels of endogenous IL-11Rα (#20 and #28) led to enhanced STAT3 phosphorylation and transcriptional activity and enhanced proliferation. 

Following proteomic analysis of our parental and IL-11Rα transfected cells, significant increases in proteins involved in cell proliferation, migration, and invasion in the high IL-11Rα transfected cells (Appendix A) led us to hypothesise that these cells would, in fact, proliferate, migrate, and invade faster than the parental cells. Indeed, both IL-11Rα-transfected cell lines significantly increased their proliferation over three days. Consistently, knocking down IL-11Rα using siRNA reduced the proliferation of these cells. These results are consistent with previous findings in other cancers that identify IL-11 signalling as a stimulator of malignant cell proliferation. For example, Wang et al. were able to attribute the recurrence of HCC in patients to the IL-11/IL-11Rα/STAT3 signalling pathway through its ability to drive proliferation [50]. This study also demonstrated that inhibiting this signalling in IL-11 knock-out mice led to an increase in apoptosis and a decrease in tumour recurrence [50]. Additionally, a study conducted by Lokau and colleagues found that antibodies targeting IL-11Rα in mice reduced overall tumour growth in endometrial cancer [25]. In fact, IL-11 has been found to be essential for colon and stomach tumourigenesis by stimulating proliferation and preventing apoptosis [51]. However, a definitive pro-proliferative role of IL-11/IL-11Rα signalling has not been identified in the glioblastoma setting. A recent study found that glioblastoma-associated microglia/macrophages secrete IL-11 into the tumour microenvironment, which promotes both tumourigenicity and resistance to temozolomide [38]. Our current data also demonstrate a clear role in IL-11/IL-11Rα signalling promoting glioblastoma progression. 

We also demonstrated that increased IL-11/IL-11Rα signalling significantly correlated with both increased migration and invasion. Interestingly, in healthy tissue, IL-11 has been identified as necessary for the migration and invasion of trophoblast cells to form the blastocyst during pregnancy [52]. Furthermore, high IL-11Rα expression in cardiac fibroblasts leads to fibrosis through overgrowth, motility, and invasion of these cells [35]. In cancer, IL-11 has been shown to have a similarly invasive role, although it has not yet been identified in the context of glioblastoma [53,54]. In a study determining the role of IL-11 signalling in endometrial cancers, the addition of IL-11 resulted in a migratory increase of 50% [52]. Similarly, the density of IL-11Rα was higher in the more invasive regions of colorectal adenocarcinoma [55]. This is particularly important, considering the function of the IL-11 cytokine is entirely determined by the presence of the receptor [56]. Together with our findings, these observations suggest that cancer cells can exploit the normal process of EMT to migrate, invade, and metastasise using IL-11 signalling [53,57,58]. In addition, IL-11 stimulation led to enhanced cell motility in chondrosarcoma tumour cells, and knockdown of either IL-11Rα or GP130 in these cells significantly reduced migration [59]. IL-11 was also shown to increase proliferation, migration, and invasion in osteosarcoma [47,48,49]. In fact, in a study conducted by Lewis and co-workers, targeting IL-11Rα using the proapoptotic agent BMTP-11 significantly decreased the overall tumourigenesis of osteosarcoma tumours [47,60]. BMTP-11 has also successfully been used to treat human leukaemia and lymphoma cell lines [60]. Unsurprisingly, IL-11Rα expression was also found to be increased in leukaemia and lymphoma, among other cancers [61,62]. Together, our findings suggest that IL-11 may be a marker of glioblastoma tumour aggressiveness with prognostic value. 

This study determined pro-tumourigenic role of IL-11 signalling in the glioblastoma setting using in vitro experimentation. Clearly, a limitation and future study extending this current dataset is to determine if these pro-tumourigenic features occur in animal xenografts or genetic models in the glioblastoma context. In addition, evaluating glioblastoma cells with varied IL-11 signaling (rather than our over or under-expressing systems used here) may also be important to conclusively identify Il-11 signaling as a critical pathway in glioblastoma pathogenesis.

Nonetheless, our current discoveries identify IL-11Rα as a possible therapeutic target for treating glioblastoma. Our findings here suggest that a therapeutic strategy that includes successful blocking of the IL-11 signaling pathway should result in reduced glioblastoma cell proliferation and invasion and thus could potentially improve patient outcomes. 

## 5. Conclusions

Overall, our discovery of IL-11Rα signalling promoting glioblastoma progression not only contributes to an improved understanding of the pro-tumour microenvironment in glioblastoma progression but also provides a potential target for therapeutic intervention through blocking the IL-11Rα signalling axis.

## Figures and Tables

**Figure 1 brainsci-14-00089-f001:**
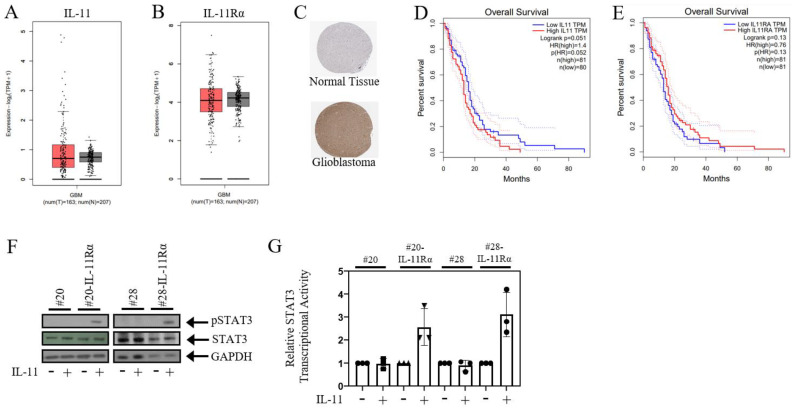
IL-11Rα signalling correlates with poorer glioblastoma patient survival. Box plots illustrate the expression levels of (**A**) IL-11 and (**B**) IL-11Rα in glioblastoma (red) and normal (grey) tissue. (**C**) Representative immunohistochemistry images of IL-11Rα in glioblastoma and normal brain tissues derived from the HPA database. The relationship between (**D**) IL-11 and (**E**) IL-11Rα gene expression and overall patient survival was determined using the TCGA database, with both IL-11 and IL-11Rα gene expression correlated with poorer overall patient survival (*p* < 0.05). (**F**) Cells were treated with ±IL-11 for 1 h and then assessed for phospho-STAT3, STAT3, and GAPDH expression by Western blot. (**G**) Cells were infected with the *Ad-APRE-luc* adenovirus and, after 24 h, treated with ±IL-11 for a subsequent 24 h. Cells were then lysed and assessed for relative STAT3 transcriptional activity, where untreated cells were normalised to 1.

**Figure 2 brainsci-14-00089-f002:**
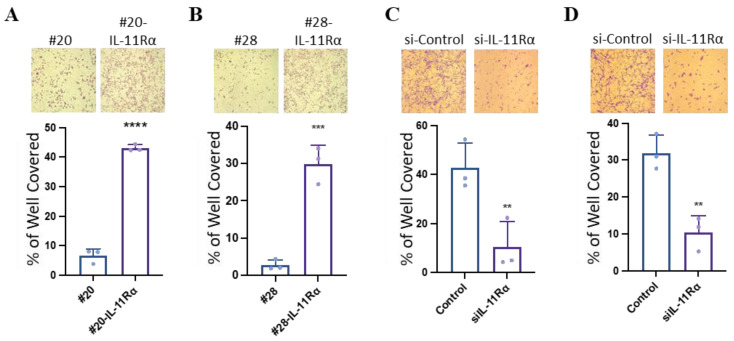
IL-11Rα expression promotes glioblastoma cell proliferation. (**A**) #20 and #20-IL-11Rα cells and (**B**) #28 and #28-IL-11Rα cells were cultured in DME media containing FBS (5%) for three days, then assessed for proliferation using the cell viability assay (n = 3, mean ± SD, where *** indicates *p* < 0.001 and **** indicates *p* < 0.0001; Mag = 50×). (**C**) #20-IL-11Rα and (**D**) #28-IL-11Rα cells were transfected with control or IL-11Rα siRNA and then cultured in DME media containing FBS (5%) for three days, then assessed for proliferation using the cell viability assay (n = 3, mean ± SD, where ** indicates *p* < 0.01; Mag = 50×).

**Figure 3 brainsci-14-00089-f003:**
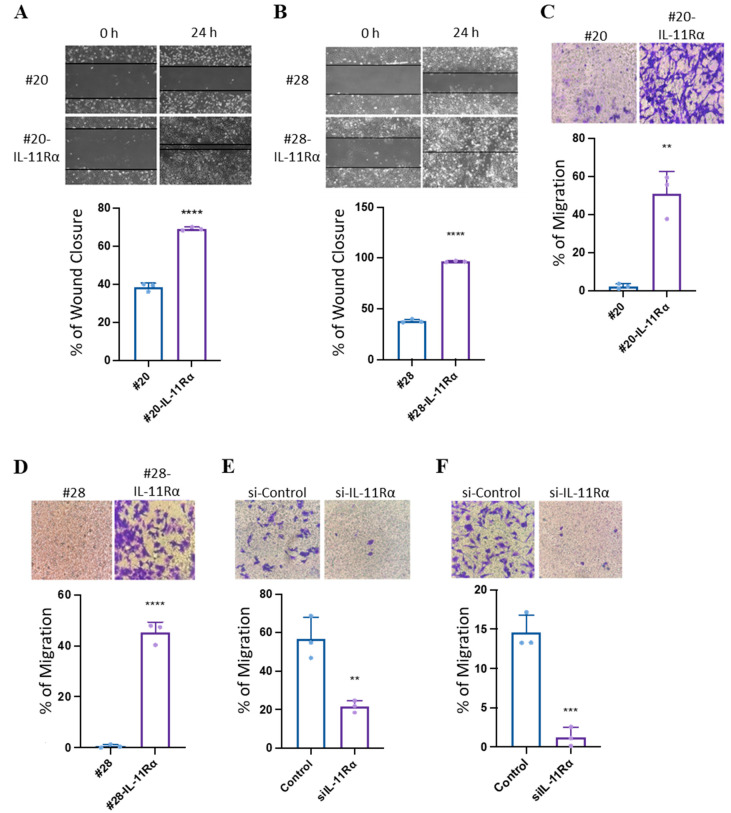
IL-11Rα expression promotes glioblastoma cell migration. (**A**) #20 and #20-IL-11Rα cells and (**B**) #28 and #28-IL-11Rα cells were cultured in DME media containing FBS (5%) until they formed confluent monolayers. The monolayers were scratched and imaged at 0 and 24 h. Image software analysis was used to determine the percentage of space covered by cells at 24 h (100% = no visible scratch/total wound healed) (n = 3, mean +/− SD, where **** indicates *p* < 0.0001). (**C**) #20 and #20-IL-11Rα cells and (**D**) #28 and #28-IL-11Rα cells were seeded onto a transwell membrane and left for 24 h. The membrane was then fixed and stained with crystal violet (0.25%). Image analysis software was used to determine the percentage of cells stained by the crystal violet after 24 h (n = 3, mean +/− SD, where ** indicates *p* < 0.01; **** indicates *p* < 0.001). Representative staining depicts cell migration of cell lines after 24 h (mag = 200×). (**E**) #20-IL-11Rα and (**F**) #28-IL-11Rα cells were transfected with control or IL-11Rα siRNA and then seeded onto a transwell membrane and left for 24 h. The membrane was then fixed and stained with crystal violet (0.25%). Image analysis software was used to determine the percentage of cells stained by the crystal violet after 24 h (n = 3, mean +/− SD, where ** indicates *p* < 0.01; *** indicates *p* < 0.005). Representative staining depicts cell migration of control and siIL-11Rα transfected cells after 24 h (mag = 200×).

**Figure 4 brainsci-14-00089-f004:**
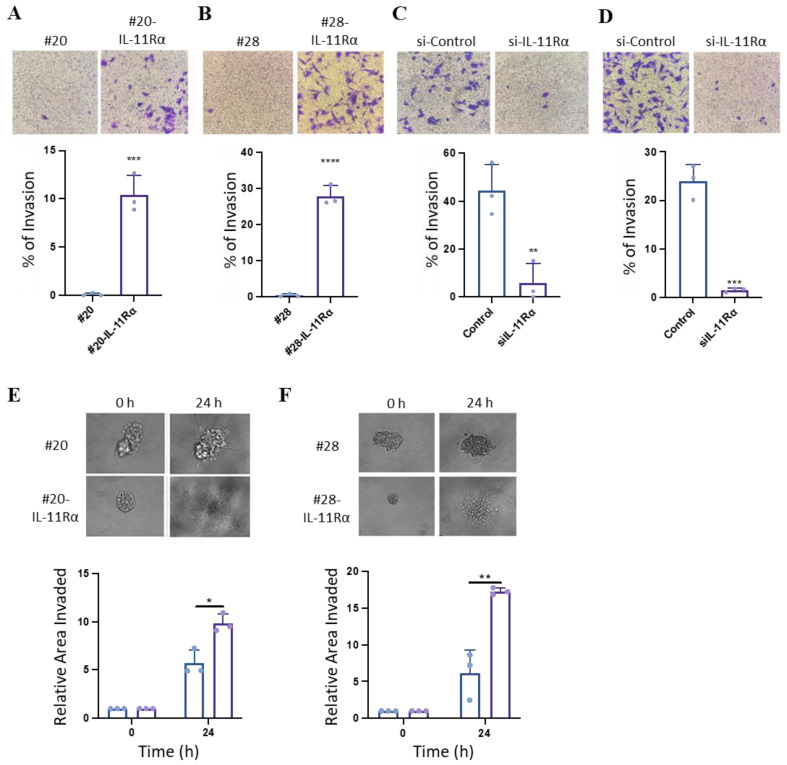
IL-11Rα expression promotes glioblastoma cell invasion. (**A**) #20 and #20-IL-11Rα and (**B**) #28 and #28-IL-11Rα cells were seeded onto a transwell membrane coated with Matrigel and left for 24 h. The membrane was then fixed and stained with crystal violet (0.25%). Image analysis software was used to determine the percentage of cells stained by the crystal violet after 24 h (n = 3, mean +/− SD, where *** indicates *p* < 0.01; **** indicates *p* < 0.001). Representative staining depicts cell invasion of cell lines after 24 h (mag = 200×). (**C**) #20-IL-11Rα and (**D**) #28-IL-11Rα cells were transfected with control or IL-11Rα siRNA and then seeded onto a transwell membrane coated with Matrigel and left for 24 h. The membrane was then fixed and stained with crystal violet (0.25%). Image analysis software was used to determine the percentage of cells stained by the crystal violet after 24 h (n = 3, mean +/− SD, where ** indicates *p* < 0.01; *** indicates *p* < 0.005). Representative staining depicts cell invasion of control and siIL-11Rα transfected cells after 24 h (mag = 200×). (**E**) #20 and #20-IL-11Rα and (**F**) #28 and #28-IL-11Rα cells were seeded onto non-adherent 24-well plates and left to grow into spheroids. The spheroids were then injected into Matrigel and imaged at 0 and 24 h. Image analysis software was used to determine the relative growth (fold change) of the cell lines over time (n = 3, mean +/− SD, where * indicates *p* < 0.05; ** indicates *p* < 0.01). Representative images depict the invasion of #20 and #20-IL-11Rα spheres at 0 and 24 h (mag = 200×).

**Figure 5 brainsci-14-00089-f005:**
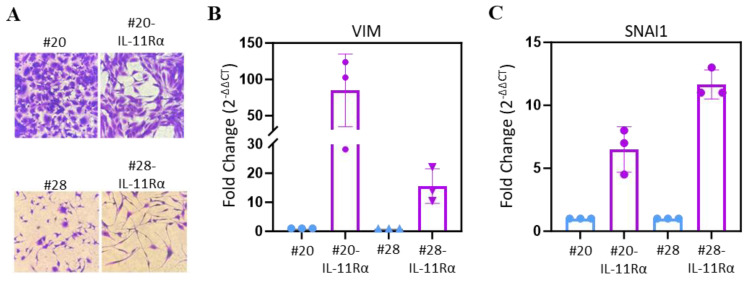
IL-11Rα expression promotes glioblastoma cell EMT properties. (**A**) #20, #20-IL-11Rα, #28, and #28-IL-11Rα cells were cultured in DME media containing FBS (5%) and allowed to adhere before being fixed with formalin and stained with crystal violet (0.25%) to observe phenotype (mag = 200×). qPCR was conducted on (**B**) #20 and #20-IL-11Rα and (**C**) #28 and #28-IL-11Rα cells to determine the gene expression of Vimentin. The results have been converted to 2^ΔΔCT^ to determine the relative fold change, in which the parental cell line was given a fold change of 1.

## Data Availability

The data presented in this study are available on request from the corresponding author. The data are not publicly available because they are not currently in a data repository.

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
