# Peer review of "Interleukin-11/IL-11 Receptor Promotes Glioblastoma Cell Proliferation, Epithelial–Mesenchymal Transition, and Invasion"

_brainsci, 2024, doi:10.3390/brainsci14010089_

Round 1
Reviewer 1 Report
Comments and Suggestions for Authors
As a brain tumor with worst prognosis, glioblastoma is highly proliferative and invasive with high fatality after the diagnosis. The immune system plays a crucial role in proliferation and expansion of glioblastoma, however, the role of cytokines responsible for promoting glioblastoma proliferation and invasion are not fully elucidated. In this study, the authors report that elevated enhanced IL-11/IL-11Rα expression was associated with reduced overall survival in glioblastoma patients. Glioblastoma cells over-expressing IL-11Rα had proteomic makeup favouring proliferation and invasion. Furthermore, knockdown of IL-11Rα resulted in reduced invasion and IL-11Rα over-expressing resulted in mesenchymal-like phenotype.
This is an interesting study on a serious pathology which can add to the body of our knowledge. However, there are some comments which need to be addressed before the manuscript is considered further. I provide detailed comments to help the authors improve their manuscript.
1. Abstract: Was this from cell line or biopsy from resected brain tissue? “We identified enhanced IL-11/IL-11Rα expression correlated with reduced overall survival in glioblastoma patients.”
2. Introduction: Some more background information on prevalence and prognosis of glioblastoma should be provided.
3. Please consider mentioning that glioblastoma can have wider implication, for example increased risk of suicidal ideation and attempt to broaden the scope of introduction. The following paper can be cited: Suicidal ideation and attempts in brain tumor patients and survivors: A systematic review. PMID: 37313501
4. Introduction: This sentence needs a reference: “Specifically, IL-11 has been shown to increase tumour proliferation, migration, invasion and survival, all important hallmarks of cancer.”
5. Line 68: “Taken together, our experimental findings suggest IL-11 signalling plays a potential prognostic role in glioblastoma and that IL-11Rα expression increases glioblastoma cell proliferation, migration and invasion.” Please do not mention results in the introduction and leave it to the Results section.
6. The Methods section is very detailed and thorough.
7. Results: Line 195: It is not clear what is meant by this: “Here, we firstly extended this analysis using additional publicly available databases.”
8. Figures: all microscopy images should include a scale bar.
9. Line 199: Data about the patients are mentioned, but no mention of patients is present in the Methods section. This is an important caveat. “Both IL-11 and Il-11Rα gene expression significantly correlated with reduced overall survival of glioblastoma patients (Fig 1D, E),”
10. Figure 1D and 1E: what is the source of this patient population?
11. Figure 1G: please include individual datapoints similar to other graphs
12. Figures 3A and 3B have different exposure in my opinion. Please elaborate.
13. Results: Line 237: Was there any staining of cell adhesion markers? “As IL-11Rα could confer enhanced proliferation, we next examined if IL-11Rα could also promote glioblastoma cell migration and invasion”
14. Supplementary Table 1: What are tick signs in the columns for proliferation and migration? This is not clear.
15. Line 282: Such sentences need quantification because this can be just sampling bias: “Visually, both 20-IL-11Rα and 28-Il-11Rα cells displayed a more mesenchymal-like morphology compared to their parental counterparts”
16. Discussion: a paragraph is needed at the end of discussion mentioning limitations of the study
17. Discussion: Please clarify the therapeutic importance of these findings.
18. Line 335: What type of cancer? This is not clear.
19. Conclusion is brief and needs to be expanded.
Comments on the Quality of English LanguageSome minor editing is required.
Reviewer 2 Report
Comments and Suggestions for Authors
In the manuscript by Stuart et al., the authors demonstrate a role of IL-11 and IL-11R in glioblastoma cell proliferation, EMT, and invasion. Most data is based on primary cell culture data, although correlative human data is provided by TCGA database exploration. Overall, the study is generally well performed and the manuscript well written.
My primary concern is related to the use of the TCGA database. It should be clearly noted whether or not all "GBM" cases that were selected are IDH-wildtype. The TCGA database was built when terms like IDH-mutant glioblastoma were used; however, glioblastoma is now considered IDH-wildtype by definition.
Additionally, vimentin is a rather non-specific marker of mesenchymal phenotype. At least one additional mesenchymal marker should be shown to better support the notion that IL-11Ralpha expression promotes GBM EMT. It would also be good to see protein expression of mesenchymal markers, not just RNA expression.
Comments on the Quality of English LanguageAs above.
Round 2
Reviewer 1 Report
Comments and Suggestions for Authors
The authors have addressed all my comments well.